



# Subgrid snow depth coefficient of variation within complex mountainous terrain

Graham A. Sexstone[1,*], Steven R. Fassnacht[2,3,4], Juan Ignacio López-Moreno[5], Christopher A. Hiemstra[6]

[1] EASC-Watershed Science, Colorado State University, Fort Collins, Colorado 80523-1476, USA
[2] ESS-Watershed Science, Colorado State University, Fort Collins, Colorado 80523-1476, USA
[3] Cooperative Institute for Research in the Atmosphere, Fort Collins, Colorado 80523-1375, USA
[4] Geospatial Centroid, Colorado State University, Fort Collins, Colorado 80523-1476, USA
[5] Instituto Pirenaico de Ecología, CSIC, Campus de Aula Dei, P.O. Box 202, Zaragoza, 50080, Spain
[6] U.S. Army Cold Regions Research and Engineering Laboratory, Fort Wainwright, Alaska 99703-0170, USA

*Correspondence to*: Graham A. Sexstone (graham.sexstone@colostate.edu)

**Abstract.** Given the substantial variability of snow in complex mountainous terrain, a considerable challenge of coarse scale modeling applications is accurately representing the subgrid variability of snowpack properties. The snow depth coefficient of variation ($CV_{ds}$) is a useful metric for characterizing subgrid snow distributions but has not

been well defined by a parameterization for mountainous environments. This study utilizes lidar-derived snow depth datasets from mountainous terrain in Colorado, USA to evaluate the variability of subgrid snow distributions within a grid size comparable to a 1000 m resolution common for hydrologic and land surface models. The subgrid $CV_{ds}$ exhibited a wide range of variability across the 321 km$^2$ study area (0.15 to 2.74) and was significantly greater in alpine areas compared to subalpine areas. Mean snow depth was an important factor of $CV_{ds}$ variability in both

alpine and subalpine areas, as $CV_{ds}$ decreased nonlinearly with increasing snow depths. This negative correlation is attributed to the static size of roughness elements (topography and canopy) that strongly influences seasonal snow variability. Subgrid $CV_{ds}$ was also correlated with topography and forest variables; important drivers of $CV_{ds}$ included the subgrid variability of terrain exposure to wind in alpine areas and the mean and variability of forest metrics in subalpine areas. Two simple statistical models were developed (alpine and subalpine) for predicting

subgrid $CV_{ds}$ that show reasonable performance statistics. The methodology presented here can be used for parameterizing $CV_{ds}$ in snow-dominated mountainous regions, and highlights the utility of using lidar-derived snow datasets for improving model representations of snow processes.

## 1 Introduction

Snow plays an important role in hydrological, ecological, and atmospheric processes within much of the Earth

System, and for this reason, considerable research has focused on understanding the spatial and temporal distribution of snow depth ($d_s$) and snow water equivalent (*SWE*) across the landscape (e.g., Clark et al., 2011). Snowpacks exhibit substantial spatiotemporal variability (e.g., López-Moreno et al., 2015) that is shaped by processes at varying spatial scales (Blöschl, 1999). The variability of the snowpack through space and time at a given scale of interest is often driven by meteorology and its interactions with topography and forest features. Mountainous areas, which

often accumulate large seasonal snowpacks, generally exhibit a high range of snow variability because of these



effects (Sturm et al., 1995). Given that this variability occurs over relatively short distances (e.g., Fassnacht and Deems, 2006; López-Moreno et al., 2011), accurately modeling the distribution of snow in mountainous areas requires a detailed understanding of the characteristics of snow variability at the model scale of interest (Trujillo and Lehning, 2015).

5    An important challenge of physically-based modeling is often the ability to represent within grid processes, or the subgrid spatial variability, of critical input parameters (Seyfried and Wilcox, 1995). Accurate representation of subgrid snow distribution is critical for reliably simulating energy and mass exchanges between the land and atmosphere in snow-covered regions (Liston, 1999), yet various studies have highlighted a deficiency with this representation in hydrologic and land-surface models (e.g., Clark et al., 2011; Liston, 2004; Liston and Hiemstra, 10    2011; Pomeroy et al., 1998; Slater et al., 2001). Liston (2004) presented an approach of effectively representing subgrid snow distributions in coarse-scale models by using a lognormal probability density function and an assigned coefficient of variation ($CV$). This approach only requires an estimation of the $CV$ parameter (i.e. standard deviation divided by the mean), which has generally been estimated from field data and is a measure of snow variability that allows for comparisons that are independent of the amount of snow accumulation. Representative values of the $CV$ 15    of snow water equivalent ($CV_{SWE}$) and snow depth ($CV_{ds}$) have been published by many field studies (refer to Table 1 and Figure 2 from Clark et al., 2011) and have been summarized based on vegetation and landform type (Pomeroy et al., 1998) and classified globally, based on air temperature, topography, and wind speed regimes (Liston, 2004). However, the range of published $CV_{SWE}$ and $CV_{ds}$ in complex mountainous terrain (i.e. the mountain snow class from Sturm et al. (1995)) is quite variable and a parameterization has not been well defined.

20    The recent developments of snow depth mapping capabilities from ground-based and airborne lidar (e.g., Deems et al., 2013) as well as digital photogrammetry (e.g., Bühler et al., 2015; Nolan et al., 2015) have provided a high definition view of snow depth distributions, albeit at fixed locations in space and time, that have not been historically available by traditional field measurements. These detailed snow depth datasets have aided in an improved understanding of the scaling properties of snow distributions (e.g., Deems et al., 2006; Trujillo et al., 25    2007), the temporal evolution of snow distributions (e.g., Grünewald et al., 2010; López-Moreno et al., 2015), the relation of snow depth with topography (e.g., Grünewald et al., 2013; Kirchner et al., 2014; Revuelto et al., 2014) and canopy (e.g., Broxton et al., 2015; Revuelto et al., 2015; Zheng et al., 2016) characteristics, as well as the nature of fine scale subgrid variability of snow depth (López-Moreno et al., 2015). Grünewald et al. (2013) present a novel study in which lidar-derived snow depth datasets are aggregated to coarse scale grids to evaluate the drivers of snow 30    distribution at the catchment scale. Evaluations of lidar snow depth datasets within coarser scale grid resolutions can be analogous to the grid resolution of many modeling applications, thus lidar-derived snow datasets have potential to serve as an important tool for evaluating the representation of subgrid snow distributions within physically-based models.

In this study, we use the snow depth coefficient of variation ($CV_{ds}$) as a metric of subgrid snow variability 35    within complex mountainous terrain similarly to López-Moreno et al. (2015), however we use a grid size comparable to a 1000 m resolution common for hydrologic and land surface models. The objectives of this research were to (1) determine the range of $CV_{ds}$ values that are observed within varying grid resolutions throughout the study





area, (2) evaluate the effects of mean snow depth, forest, and topography characteristics on subgrid $CV_{ds}$, and (3) develop a methodology for parameterizing $CV_{ds}$ within complex mountainous terrain. This research aims to help advance understanding of the variability of subgrid snow distributions, and support the development of more accurate representations of subgrid snow variability that can be used within physically-based models.

## 2 Methods

### 2.1 Site description

The study area is in the Front Range Mountains of north-central Colorado, located in the western United States (Figure 1). Spatial lidar datasets collected by the Boulder Creek Critical Zone Observatory (CZO) (http://criticalzone.org/boulder/, accessed 17 April 2016) were investigated in this study. The study area (321 km$^2$) ranges in elevation from 2190 m to 4117 m and is dominated by ponderosa pine (*Pinus ponderosa*) and lodgepole pine (*Pinus contorta*) at lower elevations, Engelmann spruce (*Picea engelmannii*) and subalpine fir (*Abies lasiocarpa*) forests at higher elevations, and alpine tundra at the highest elevations (Figure 1). The mean winter (1 October to 1 May) precipitation and temperature for water years 2006 - 2010 at the Niwot SNOTEL site (3021 m; Figure 1) is 452 mm and 2.7°C (Harpold et al., 2014). The mountainous terrain in this region is complex, varying from gentle topography at lower elevations to steep and rugged slopes closer to the Continental Divide. The majority of the study area has a southeastern aspect and is located on the eastern side of the Continental Divide (Figure 1). The Front Range Mountains are characterized by a continental seasonal snowpack (Trujillo and Molotch, 2014), with the persistent snow zone at elevations greater than 3050 m (Richer et al., 2013), generally exhibiting peak snow accumulation during the springtime months of April and May each year.

### 2.2 Spatial datasets

This analysis uses publically available lidar-derived snow depth ($d_s$), elevation ($z$), and vegetation height (*VH*) raster datasets (1 m resolution) from the Boulder Creek CZO (ftp://snowserver.colorado.edu/pub/ WesternCZO_LiDAR_data, accessed 27 August 2015) that are described in detail by Harpold et al. (2014). Airborne lidar campaigns were completed during snow-covered (May 2010) and snow free (August 2010) periods across the study area and lidar surfaces were differenced to derive $d_s$ (Harpold et al., 2014). The snow-covered lidar returns were collected on two dates, 05 May 2010 and 20 May 2010, and the combined snow-covered lidar extent is 321 km$^2$ (Figure 1). Snow-covered lidar collections were carried out during this time of the year to capture the near-peak snow accumulation (Harpold et al., 2014), and the two survey dates (15 days apart) required to cover the study area extent were a result of weather challenges and equipment failures (http://czo.colorado.edu/geGIS/ 0README_BcCZO_LiDAR.pdf, accessed 02 September 2016). Given that studies have observed inter-annual consistency in snow accumulation patters (e.g., Deems et al., 2008; Erickson et al., 2005; Sturm and Wagner, 2010), we expect the 2010 lidar surveys to be somewhat representative of snow variability and $CV_{ds}$ near peak snow accumulation across the study area. A comparison of the lidar $d_s$ dataset to *in-situ* $d_s$ sensors within research





catchments in the Boulder Creek CZO showed a root mean squared error (RMSE) of 27 cm and 7 cm at the Como Creek catchment (16 sensors) and Gordon Gulch catchment (5 sensors), respectively (Harpold et al., 2014).

The lidar-derived digital elevation model (DEM) was resampled from a 1 m to a 10 m resolution for representation of the resolution of commonly available DEMs (USGS National Elevation Dataset, http://ned.usgs.gov) and was subsequently used to derive topography variables for each 10 m cell that have been
shown to influence $d_s$ distributions (e.g., Elder et al., 1998; Erickson et al., 2005; Kerr et al., 2013; Revuelto et al., 2014; Winstral et al., 2002) using a Geographic Information System (GIS). Surface slope ($S$) was calculated by fitting a plane to a 3 x 3 cell window around each DEM cell. Winter clear-sky incoming solar radiation ($Q_{sw\downarrow}$) was determined using the Area Solar Radiation tool in ArcGIS, which calculates mean incoming solar radiation for clear-
sky conditions across a DEM surface for a specified time interval based on solar zenith angle and terrain shading. The time interval used for the calculation of $Q_{sw\downarrow}$ was 01 October through 01 May. Aspect was not considered because it was highly correlated with $Q_{sw\downarrow}$. Maximum upwind slope ($Sx$) (Winstral et al., 2002), which can be used as a measure of the exposure to or sheltering from wind, was calculated for each cell as:

$$Sx_{\alpha,d\,max}\left(x_i,y_i\right)=\max\left(\tan^{-1}\left\{\frac{z\left(x_v,y_v\right)-z\left(x_i,y_i\right)}{\left[\left(x_v-x_i\right)^2+\left(y_v-y_i\right)^2\right]^{0.5}}\right\}\right) \qquad (1)$$

where $\alpha$ is the azimuth of the search direction, $dmax$ is the maximum distance for the search direction, $z$ is elevation, and ($x_v$, $y_v$) are all cells along the vector defined by $\alpha$ and $dmax$. Given the prevailing westerly winds within the study area (Erickson et al., 2005; Winstral et al., 2002), an average $Sx$ was calculated for a $dmax$ of 200 m and a range of $\alpha$ from 240° to 300° at 5° increments (e.g., Molotch et al., 2005). Topographic position index ($TPI$) (Weiss, 2001), which is a measure of the relative position of the cell to surrounding topography, was calculated for each cell
as:

$$TPI = z_0 - \bar{z} \qquad (2)$$

$$\bar{z} = \frac{1}{n_R}\sum_{i\in R} z_i \qquad (3)$$

where $z_0$ is the elevation of the cell and $\bar{z}$ is the average elevation of the surrounding cells within a specified cell window ($R$). TPI was calculated for 3 x 3 (i.e. 30 m resolution), 11 x 11, and 21 x 21 windows around each cell.

Additional forest canopy spatial datasets were also used in this study. WorldView-2 (WV2) satellite imagery (DigitalGlobe, Inc., USA) from a cloud free sky condition on 26 September 2013 was acquired. The WV2 imagery has a high spatial (3 m) and spectral resolution (8 multispectral bands) and was used to compute the Normalized Difference Vegetation Index ($NDVI$) for the study area at a 3 m resolution. Additionally, a 30 m resolution 2011 canopy density ($CD$) dataset was acquired for the study area (http://www.mrlc.gov/nlcd2011.php,
accessed 04 December 2015).

## 2.3 Aggregation of study grids

Operational snow models (e.g., Carroll et al., 2006) often have a 1000 m horizontal grid resolution and snow representations within land surface models (e.g., Slater et al., 2001) have generally been designed for a coarser



resolution (e.g., Yang et al., 1997) but are being developed to operate at finer scales (e.g., Bierkens et al., 2015; Kumar et al., 2006; Wood et al., 2011). This study attempts to evaluate the subgrid variability of $d_s$ at a comparable grid resolution to this 1000 m model grid size. The study area was automatically divided into square grids of equal size (Figure 1), hereinafter referred to as study grids, and the subgrid variability of $d_s$ was computed for each study grid. We tested a range of study grid sizes including 100 m, 200 m, 300 m, 400 m, 500 m, 750 m, and 1000 m to determine how similar the subgrid $d_s$ characteristics of each of the grid sizes were to that of 1000 m. Although the direct evaluation of subgrid snow variability at the 1000 m grid size was provided, the use of a smaller grid size with similar characteristics to 1000 m grid size provides a greater number of grids for statistical analysis (i.e. greater sample size). Therefore, the goal of this testing was to identify an appropriate grid size for evaluation that exhibited similar characteristics of snow variability to the 1000 m resolution grids, but maximized the number of grids available for analysis within the study area. Study grids were required to have at least 80% coverage by the lidar $d_s$ datasets, and the $d_s$ dataset with the greatest coverage was utilized for cases of the overlapping snow-covered lidar datasets (Figure 1). When the 05 May 2010 and 20 May 2010 lidar $d_s$ datasets were overlapping and both datasets had 100% study grid coverage, the 05 May 2010 dataset was used. In order to assess the influence of using lidar-derived snow depth from two different days, the snow depth distributions within the overlapping area of the two lidar campaigns (7.92 km$^2$; Figure 1) were compared.

For each study grid, the mean and standard deviation of $d_s$ were determined and used to calculate $CV_{ds}$. The mean and standard deviation of each of the topography and vegetation datasets outlined above were also calculated for each study grid. A categorical variable representing ecosystem type was also determined for each study grid. The alpine ecosystem type was assigned to study grids that had a mean elevation greater than 3300 m and a mean $VH$ less than 0.5 m, while the remaining study grids were assigned to the subalpine ecosystem type (Figure 1); treeline elevation in this area generally varies between 3400 m and 3700 m (Suding et al., 2015). Lastly, only study grids with a mean elevation greater than 3000 m (i.e. the persistent snow zone) were evaluated in this study (Figure 1).

**2.4 Statistical analysis**

Pairwise relations between $CV_{ds}$ and $d_s$, topography variables (mean and standard deviation), and vegetation variables (mean and standard deviation) were explored for both alpine and subalpine study grids to evaluate drivers of subgrid $d_s$ variability. $CV_{ds}$ was expected to have a strong nonlinear relation with $d_s$ (Fassnacht and Hultstrand, 2015); therefore, the influence of $d_s$ on $CV_{ds}$ was removed (i.e. detrended) using a best-fit power function for both the alpine and subalpine study grids, and residuals were used to evaluate further topography and vegetation effects on $CV_{ds}$ using Pearson's $r$ coefficient. Additionally, multiple linear regression models were developed to predict $CV_{ds}$ for both alpine and subalpine study grids. We evaluated a range of independent variables to be included within the multiple linear regression models (refer to variables in Table 1). However, given that the goal of the model analysis was to provide a methodology for parameterizing $CV_{ds}$, some of the variables were deemed unsuitable and excluded from model testing. For example, mean $z$ was not included in model testing as it was believed to be a site specific variable that may not have been transferable to independent data. Additionally, $VH$ was not tested within the models as spatial datasets of this variable are not commonly available, unlike the USGS National Land Cover



Database (http://www.mrlc.gov/index.php) canopy density product or remote sensing forest metrics such as *NDVI*. Variables included in the models were selected by an all-subsets regression procedure in which both Mallows' $C_p$ (Mallows, 1973) and Akaike information criterion (AIC) (Akaike, 1974) were used as a measure of relative goodness of fit of the models (e.g., Sexstone and Fassnacht, 2014). Final independent variables within the models were required to be statistically significant ($p$ value $< 0.05$) and not exhibit multicollinearity. Multicollinearity was defined as model parameters exhibiting a variance inflation factor greater than 2. Given that a non-normal distribution of snow depth (Liston, 2004) and other topography and vegetation variables was expected, various transformations of model variables were explored. Model diagnostics of residuals were used to ensure the model assumptions of normality, linearity, and homoscedasticity. Model performance was evaluated using the Nash-Sutcliffe efficiency (NSE) and RMSE. Additionally, model verification was assessed using a 10-fold cross-verification procedure which runs 10 iterations of removing a randomly-selected 10 percent of the dataset, fitting the regression to the remainder of the data, and subsequently comparing modeled values to the independent observations that were removed.

## 3 Results

### 3.1 Snowpack conditions

In this study, an evaluation of the snowpack conditions was important for assessing if the subgrid $CV_{ds}$ may have been influenced by a melting snowpack. In a hypothetical uniform snowmelt scenario (e.g., Egli and Jonas, 2009), the subgrid mean $d_s$ is expected to decrease faster than the $\sigma_{ds}$, thus the $CV_{ds}$ will increase without a corresponding increase in subgrid snow variability (Winstral and Marks, 2014). *SWE* data from nine Natural Resources Conservation Service (NRCS) SNOTEL stations located in the Front Range Mountains of northern Colorado (Figure 1) were evaluated to assess snowpack conditions. A snowmelt event occurred across the study area on 10 April 2010 (Figure 2a) that caused considerable snowmelt at stations below an elevation of 3000 m and a loss of 10% of peak *SWE* on average at stations above 3000 m. Following this snowmelt event, substantial snow accumulation continued at SNOTEL stations above 3000 m until 17 May 2010, when the onset of snowmelt began (Figure 2a). A plot of $\sigma_{ds}$ versus mean $d_s$ among the SNOTEL stations highlights the hysteretic dynamics of accumulation and melt across the region (Egli and Jonas, 2009), and confirms that the lidar data were collected prior to and at the beginning of snowmelt across the study area (Figure 2b). Additionally, the statistical distributions of snow depth on 05 May 2010 and 20 May 2010 within the areas that were overlapped by both lidar campaigns (7.92 km$^2$; Figure 1) are shown to be similar and have a $CV_{ds}$ of 1.01 and 1.10, respectively (Figure 3). Given that the lidar-derived snow depths were collected before substantial snowmelt had occurred within the persistent snow zone and the distributions of $d_s$ from both dates exhibit similar characteristics, we are confident that the subgrid $CV_{ds}$ evaluated in this study is representative of snow variability at peak snow accumulation and was not significantly influenced by data collection on two separate dates.



### 3.2 Subgrid snow depth variability

Snow depth $CV$ ($CV_{ds}$) and $\sigma_{ds}$ were consistently greater in the alpine versus subalpine at each of the varying grid resolution sizes (Figure 4). The mean $CV_{ds}$ across the study grids was generally consistent with changes in grid resolution; however, the standard deviation of $CV_{ds}$ decreased with increasing grid resolution and stabilized around a

500 m grid size. The mean $\sigma_{ds}$ across the study grids tended to increase with increasing grid size for all study grids, but stabilized around 400 m for subalpine study grids only. The 500 m resolution study grids (n = 642) were the smallest grid size with a comparable mean and standard deviation of $CV_{ds}$ to the 1000 m grid size (Figure 4) and were chosen for analysis in this study (Figure 1) as a grid size representative of the subgrid snow variability at the 1000 m resolution.

10         The median $d_s$, $\sigma_{ds}$, and $CV_{ds}$ across all study grids (hereinafter 500 m resolution) was equal to 1.27 m, 0.88 m, and 0.74, respectively, and subgrid $CV_{ds}$ ranged from 0.15 to 2.74 across the study area. The variability of $CV_{ds}$ collected on 05 May 2010 (n = 219 study grids) and 20 May 2010 (n = 423 study grids) (Figure 1) was similar, with the 05 May grids exhibiting a slightly smaller $CV_{ds}$ (median = 0.64) than the 20 May grids (median = 0.81). Statistically significant differences ($p$ value < 0.001) between the alpine and subalpine study grids were observed for

$d_s$, $\sigma_{ds}$, and $CV_{ds}$ by the nonparametric Mann-Whitney test (Figure 5). The alpine study grids exhibited a greater mean and range of snow accumulation and variability than the subalpine study grids. The range of $CV_{ds}$ from the 10th to the 90th percentiles within the alpine and subalpine study grids was equal to 0.61 to 1.57 and 0.30 to 0.98, respectively. Figure 6 highlights the abrupt change of subgrid snow depth variability characteristics observed in a transition from the subalpine to alpine ecosystem; forest structure and topography characteristics appears to exert a

strong influence on subgrid $CV_{ds}$ and these relations were investigated further.

### 3.3 Relation of subgrid snow depth variability with topography and forest characteristics

A statistically significant linear correlation (Pearson's r coefficient; $p$ value < 0.05) between $CV_{ds}$ and $d_s$ was observed to be -0.60 and -0.45 for the alpine and subalpine study grids, respectively (Table 1). However, further evaluation showed this relation to be nonlinear and best described by a power function (Figure 7). This function

suggests that $CV_{ds}$ exhibits a systematic decrease with increasing $d_s$ and suggests that relative subgrid snow variability is importantly related to the total snow accumulation of a given year. The power relation between $CV_{ds}$ and $d_s$ was greatly improved when split between alpine and subalpine study grids, as a $CV_{ds}$ for a corresponding $d_s$ tended to be greater for alpine versus subalpine study grids (Figure 7). The power functions ($CV_{ds}$ versus $d_s$) were detrended (i.e. removing the influence $d_s$ on $CV_{ds}$) and the residuals of the functions were compared to topography

and forest characteristics (Table 2). The alpine study grids were most positively correlated with $\sigma_{Sx}$ suggesting that the variability of wind exposure and sheltering and thus wind redistribution within a study grid is a strong control on $CV_{ds}$. The $\sigma_S$, $\sigma_{TPI}$, $\sigma_z$ were also closely related to $CV_{ds}$ in both alpine and subalpine areas highlighting the overall importance of topographic roughness on subgrid snow variability. The subalpine study grids were negatively correlated with the $VH$, $NDVI$, and $CD$ variables and also positively correlated with the variability of these

vegetation metrics ($CV_{VH}$, $\sigma_{NDVI}$, $\sigma_{CD}$), suggesting that forest structure is important driver of subalpine subgrid variability.





### 3.4 Statistical models

The multiple linear regression models developed for predicting $CV_{ds}$ in both alpine and subalpine seasonal snowpacks are presented in Table 2. Variable transformations were necessary to $CV_{ds}$ and $d_s$ in both models and to $\sigma_{Sx}$ in the alpine model and $CD$ in the subalpine model to account for the nonlinearity of these datasets (Table 2).

Snow depth exhibited the greatest explanatory ability within both the alpine and subalpine models, with standardized regression coefficients equal to -0.92 and -0.95, respectively (not shown). Standardized regression coefficients of $\sigma_{Sx}$ and $CD$ were equal to 0.50 and -0.72 for the alpine and subalpine models, respectively, and both showed the second strongest explanatory power in their respective models. The alpine model had a NSE of 0.66 (0.65) and RMSE of 0.24 (0.24) while the subalpine model had an NSE of 0.79 (0.78) and RMSE of 0.12 (0.13) for

the model calibration (10-fold cross-verification) dataset (Figure 8). A total NSE of 0.81 was calculated for the entire dataset based on predictions from both models. These performance statistics suggest that the models perform reasonably well predicting $CV_{ds}$ and cross-verification suggests the model may be transferable to independent data within the bounds of the original dataset.

### 4 Discussion

Based on an evaluation of $CV_{ds}$ at a 500 m grid resolution, subgrid snow variability across a mountainous subalpine and alpine study area is shown to exhibit a wide range of spatial variation and be well correlated with ecosystem type, snow amount, as well as topography characteristics and forest structure. Alpine $CV_{ds}$ was most correlated with mean snow depth and the variability of exposure to wind while mean snow depth and canopy height and density were most correlated with $CV_{ds}$ in subalpine areas. A simple statistical model for both alpine and subalpine

ecosystems was able to reasonably predict subgrid $CV_{ds}$ based on these relations and could be used as a methodology for improving model parameterizations of subgrid snow variability in mountainous terrain.

The range of $CV_{ds}$ observed over relatively small distances in this study (Figure 6) highlights the importance of further characterizing the spatial variability of this parameter within mountainous terrain. The global classification of $CV_{SWE}$ defined by Liston (2004) performed well predicting the average conditions observed in this

study. Liston (2004) define the $CV_{SWE}$ of mid-latitude mountainous forest (i.e. subalpine) as 0.60 and of mid-latitude treeless mountains (i.e. alpine) as 0.85, whereas this study found a median $CV_{ds}$ of 0.55 for subalpine study grids and 1.05 for alpine study grids. However, the global classification was unable to adequately represent the range and variability of $CV_{ds}$ across the study area (Figure 5c), and the results presented herein suggests promise for an improved parameterization of $CV_{ds}$ in mountainous terrain.

Mean snow depth was well correlated with $CV_{ds}$ variability across alpine and subalpine areas within the study area. As subgrid $d_s$ increased, the $CV_{ds}$ decreased, which is a result that is consistent with previous studies at various spatial scales (e.g., Fassnacht and Deems, 2006; Fassnacht and Hultstrand, 2015; López-Moreno et al., 2015). A positive correlation was observed between $\sigma_{ds}$ and $d_s$ in alpine and subalpine areas, which had a dampening effect on this overall negative correlation between the relative subgrid variability ($CV_{ds}$) with $d_s$. The relative subgrid

variability of $d_s$ likely decreases with increasing snow accumulation because of the consistent size of the roughness



elements of topography and canopy that drive snow variability; as $d_s$ increases, the relative influence of these topography and canopy features tends to decrease (e.g., Fassnacht and Deems, 2006; López-Moreno et al., 2011; López-Moreno et al., 2015). The range of $CV_{ds}$ observed in this study (Figure 5) is similar to previous studies conducted in mountainous mid-latitude forested and alpine areas (refer to Figure 2 from Clark et al., 2011 and

references therein). Future research could further investigate $CV_{ds}$ and $d_s$ across different geographic regions and snow regimes as well as across multiple snow seasons and compare results to the functions presented in Figure 7 to better understand the dynamics and consistency of this relation. An understanding of how the subgrid variability of snow depth for a given set of topography and canopy elements scales between low and high snow years could be particularly important.

Within the alpine study grids, the variability of the exposure/sheltering from wind ($\sigma_{Sx}$) was an important driver of $CV_{ds}$. Study grids with the greatest $\sigma_{Sx}$ were generally positioned over large breaks in topography. For example, a given study grid with a large $\sigma_{Sx}$ likely contained areas with both wind exposure ($Sx < 0°$) where snow accumulation is scoured by wind and sheltering from wind ($Sx > 0°$) where preferential deposition of wind transported snow occurs. Study grids with a consistent $Sx$ showed a lower $CV_{ds}$ with greater variability observed in

sheltered grids than in exposed grids. Winstral et al. (2002) and many subsequent studies (e.g., Erickson et al., 2005; McGrath et al., 2015; Molotch et al., 2005; Revuelto et al., 2014) have highlighted this control of wind exposure on snow depth distribution in tree-less areas. The degree of importance of $\sigma_{Sx}$ for describing $CV_{ds}$ is likely variable from year-to-year, and would be expected to be well correlated with observed wind speeds (Winstral and Marks, 2014). However, in alpine areas where high wind speeds are ubiquitous, $\sigma_{Sx}$ is expected to be a consistently important

driver of subgrid snow variability.

Subgrid snow variability within subalpine study grids was well correlated with the *VH*, *NDVI,* and *CD* vegetation metrics. As mean study grid *VH*, *NDVI*, and *CD* increased, $CV_{ds}$ tended to decrease, but was also shown to be positively correlated with the variability of these metrics ($CV_{VH}$, $\sigma_{NDVI}$, $\sigma_{CD}$). Forest structure has been shown by various studies to exert a strong influence on snow variability because of a variety of physical process

interactions. Interception of snow (e.g., Hedstrom and Pomeroy, 1998) and subsequent canopy sublimation (e.g., Molotch et al., 2007; Montesi et al., 2004), influences of trees on shortwave (e.g., Ellis and Pomeroy, 2007; Musselman et al., 2012) and longwave (e.g., Pomeroy et al., 2009) radiation dynamics, and the effect of trees on wind redistribution of snow (e.g., Hiemstra et al., 2006) can each drive snow accumulation and evolution in forested areas. Broxton et al. (2015) utilized lidar-derived snow depth datasets and showed that the variability of snow depth

in subalpine forests tended to be greatest beneath the forest canopy and near the forest canopy edge and the least snow depth variability occurred in forested openings that were distant from the forest edge. Also, substantial differences in accumulated $d_s$ were observed between subcanopy areas and forest openings. The increased $CV_{ds}$ with decreasing *VH*, *NDVI*, and *CD* observed in this study can be explained by a greater occurrence of transitional areas between subcanopy areas and forest openings (i.e., forest edges) occurring in study grids with smaller mean *VH* and

*CD*. Across the study area, subalpine forest openings that spanned an entire study grid were not present; therefore, study grids with consistent forest cover tended to exhibit the least subgrid snow variability.



This study was limited by the spatial and temporal coverage of the lidar-derived snow datasets that were used (Figure 1). Although the alpine and subalpine areas evaluated are representative of mountainous terrain in the region and snowpacks in this area are representative of the continental snow regime (Trujillo and Molotch, 2014), further analysis of subgrid snow variability across a greater geographic area and across other regions with differing

snow regimes could improve the applicability of a $CV_{ds}$ parameterization for snow distributions in mountains areas in general. Additionally, spatial patterns of snow variability have been shown to be temporally consistent from year-to-year (e.g., Deems et al., 2008; Erickson et al., 2005; Sturm and Wagner, 2010), but future studies with multiple years of lidar collection could help understand the inter-annual variability of $CV_{ds}$ and the consistency of its driving variables (e.g., Fassnacht et al., 2012). Of particular interest would be the temporal consistency of the relation

between $CV_{ds}$ and $d_s$.

This study evaluates the subgrid variability of $d_s$, but $SWE$ is the most fundamental snowpack variable of interest in land surface processes (e.g., Sturm et al., 2010). Snow depth and $SWE$ have been shown by many studies to be well correlated (e.g., Jonas et al., 2009; Sexstone and Fassnacht, 2014; Sturm et al., 2010), and the subgrid $CV$ of these variables is expected to exhibit similar characteristics (e.g., Fassnacht and Hultstrand, 2015). We suggest

that a parameterization of $CV_{ds}$ could be sufficient for representing subgrid $SWE$ variability, but further investigation into this hypothesis is needed. In order to directly investigate $CV_{SWE}$ from lidar-derived snow data in future studies, an estimation of snow density would be needed. Statistically-derived snow density models have been successfully developed over varying domain sizes for estimating $SWE$ from $d_s$ (e.g., Jonas et al., 2009; Sexstone and Fassnacht, 2014; Sturm et al., 2010), and these models make use of the fact that $SWE$ and $d_s$ variability is much greater than the

variability of snow density (e.g., Lopez-Moreno et al., 2013; Mizukami and Perica, 2008).

The snow distributions and variability characteristics evaluated in this study were likely somewhat influenced by the occurrence of snowmelt conditions within the study area. Although substantial snowmelt had not occurred prior to data collection within the study grids (Figure 2), the mid-season melt events and onset of snowmelt may have caused an increase in $CV_{ds}$ (Figure 3) and this effect may have differed between the two dates of lidar-

derived $d_s$. López-Moreno et al. (2015) observed a sharp increase in $CV_{ds}$ just following the onset of snowmelt yet a fairly consistent $CV_{ds}$ for the remainder of snowmelt season. Future studies evaluating subgrid snow variability should investigate the intra-annual variability $CV_{ds}$ to further understand the seasonal evolution of this parameter.

The development of high resolution snow depth mapping from lidar has provided a unique ability for detailed snapshot views of the spatial distribution of snow in complex mountains areas. Although some key

advantages of these datasets are related to validating satellite-based remote sensing products and direct use within water resources forecasting, this study also suggests that lidar-derived snow datasets can be an important tool for the improvement of snow representations within modeling applications. Future research should utilize lidar-derived snow datasets to directly evaluate the ability of physically-based models to represent snow distributions as well as to continue to improve the representation of subgrid variability of snow. Additionally, other key snow modeling

questions such as how representative snow monitoring stations are of surrounding areas (e.g., Meromy et al., 2013; Molotch and Bales, 2005) could also be investigated further by lidar-derived snow datasets. Lastly, the analysis methods that have been developed in this study may also be useful in future studies for characterizing the subgrid



variability of other variables that can be measured remotely at a fine scale through lidar or other measurement techniques.

## 5 Conclusions

This study outlines a methodology for utilizing lidar-derived snow datasets for investigating subgrid snow depth ($d_s$)
variability and potentially improving its representation within physically-based modeling applications. Subgrid $d_s$ variability was evaluated over a range of grid sizes and it was determined that study grid $CV_{ds}$ characteristics were similar among resolutions from 500 m to 1000 m. Study grids (500 m resolution) exhibited a wide range of $CV_{ds}$ across the study area (0.15 to 2.74) and subgrid $d_s$ variability was found to be greater in alpine areas than subalpine areas. Snow depth was the most important driver of $CV_{ds}$ variability in both alpine and subalpine areas and a
systematic nonlinear decrease in $CV_{ds}$ with increasing $d_s$ was observed; the negative correlation between $CV_{ds}$ and $d_s$ is attributed to the static size of roughness elements (topography and canopy) that strongly influence seasonal snow variability. The variability of wind exposure in alpine areas as well as vegetation metrics in subalpine areas were also found to be important drivers of study grid $CV_{ds}$. Two simple statistical models were developed (alpine and subalpine) for predicting subgrid $CV_{ds}$ from mean $d_s$ and topography/canopy features that show reasonable
performance statistics and suggest this methodology can be used for parameterizing $CV_{ds}$ in snow-dominated mountainous areas. This research highlights the utility of using lidar-derived snow datasets for improving model representations of subgrid snow variability.

### Acknowledgements

This work was partially funded by the NASA Terrestrial Hydrology Program (award NNX11AQ66G 'Improved
Characterization of Snow Depth in Complex Terrain Using Satellite Lidar Altimetry' PI Michael F. Jasinski NASA GSFC). The airborne lidar data were collected in collaboration between the Boulder Creek CZO and the National Center for Airborne Laser Mapping, both of which are funded by the National Science Foundation. The lidar-derived elevation, vegetation, and snow datasets were processed and made publically available as described in Harpold et al. (2014) and were instrumental datasets for this study. Thanks to Adam Winstral for providing the code
for computing $Sx$.

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





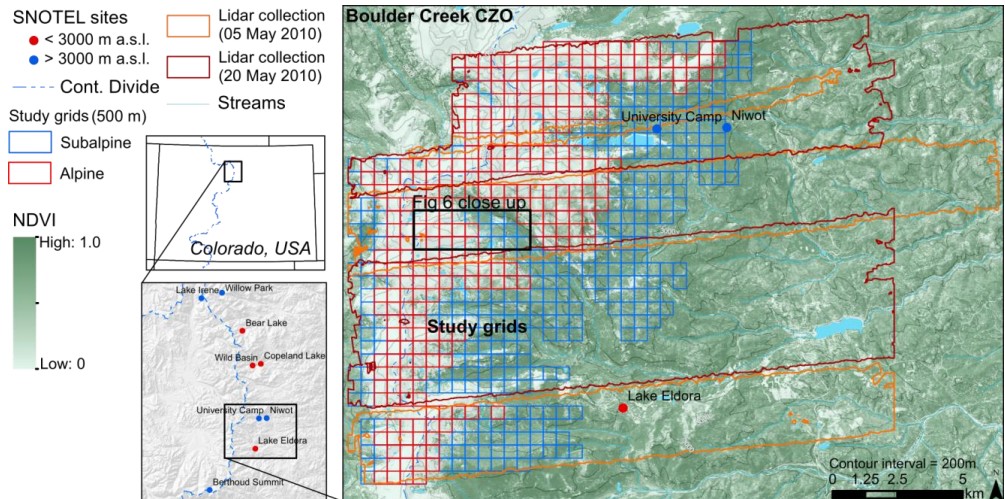

**Figure 1: Map of the Boulder Creek CZO study area located within the Front Range Mountains of northern Colorado, USA. NRCS SNOTEL sites in the region are shown in blue (sites greater than 3000 m elevation) and red (sites less than**
5 **3000 m elevation). The extent of the snow-covered lidar collection on 05 May 2010 (20 May 2010) is shown in orange (dark red). The 500 m resolution study grids (n = 650) are shown in blue (subalpine) and red (alpine). The black rectangle highlights the area of close up shown in Figure 6.**





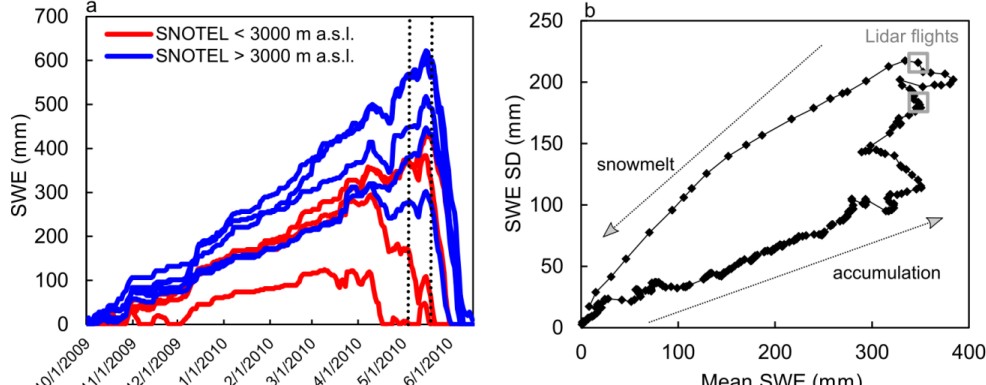

**Figure 2: Snow water equivalent (*SWE*) data from nine NRCS SNOTEL sites within the region of the study area displayed as (a) niveographs showing snow accumulation and snowmelt throughout water year 2010 with the timing of 05 May 2010 and 20 May 2010 lidar flights plotted as vertical dashed lines and (b) a scatter plot of the standard deviation of *SWE* versus mean *SWE* from the SNOTEL sites highlighting the hysteretic dynamics of snow accumulation and snowmelt across the region based on nine SNOTEL stations (Egli and Jonas, 2009).**





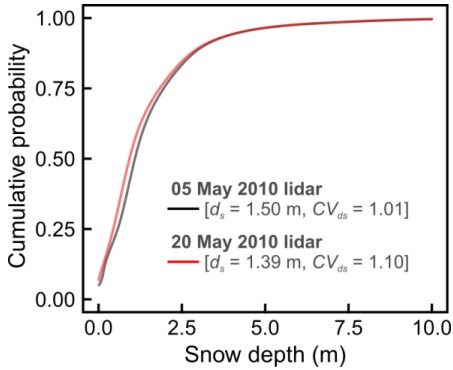

**Figure 3: Statistical distributions of lidar-derived snow depth for the overlapping area (7.92 km$^2$) of the 05 May 2010 and 20 May 2010 lidar flights.**





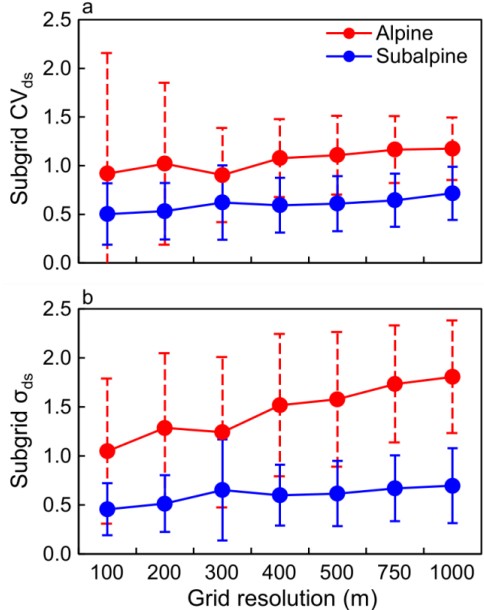

**Figure 4:** Mean subgrid (a) $CV_{ds}$ and (b) $\sigma_{ds}$ across the study area plotted versus study grid resolution for alpine (red) and subalpine (blue) study grids. Error bars represent the standard deviation of $CV_{ds}$ and $\sigma_{ds}$ across the study area.





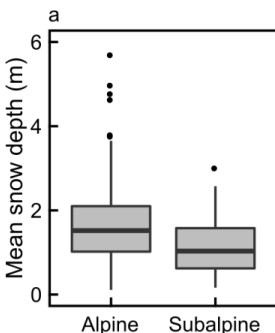 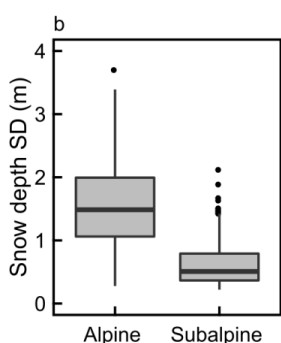 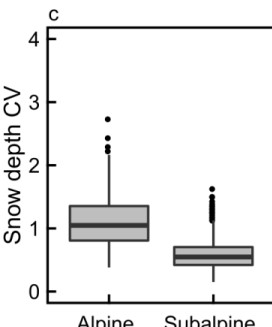

**Figure 5: Boxplots showing the outliers (black circles), 10th and 90th percentiles (whiskers), 25th and 75th percentiles (box) and median (black horizontal line) for the (a) $d_s$, (b) $\sigma_{ds}$, and (c) $CV_{ds}$ of the alpine and subalpine study grids (500 m resolution).**




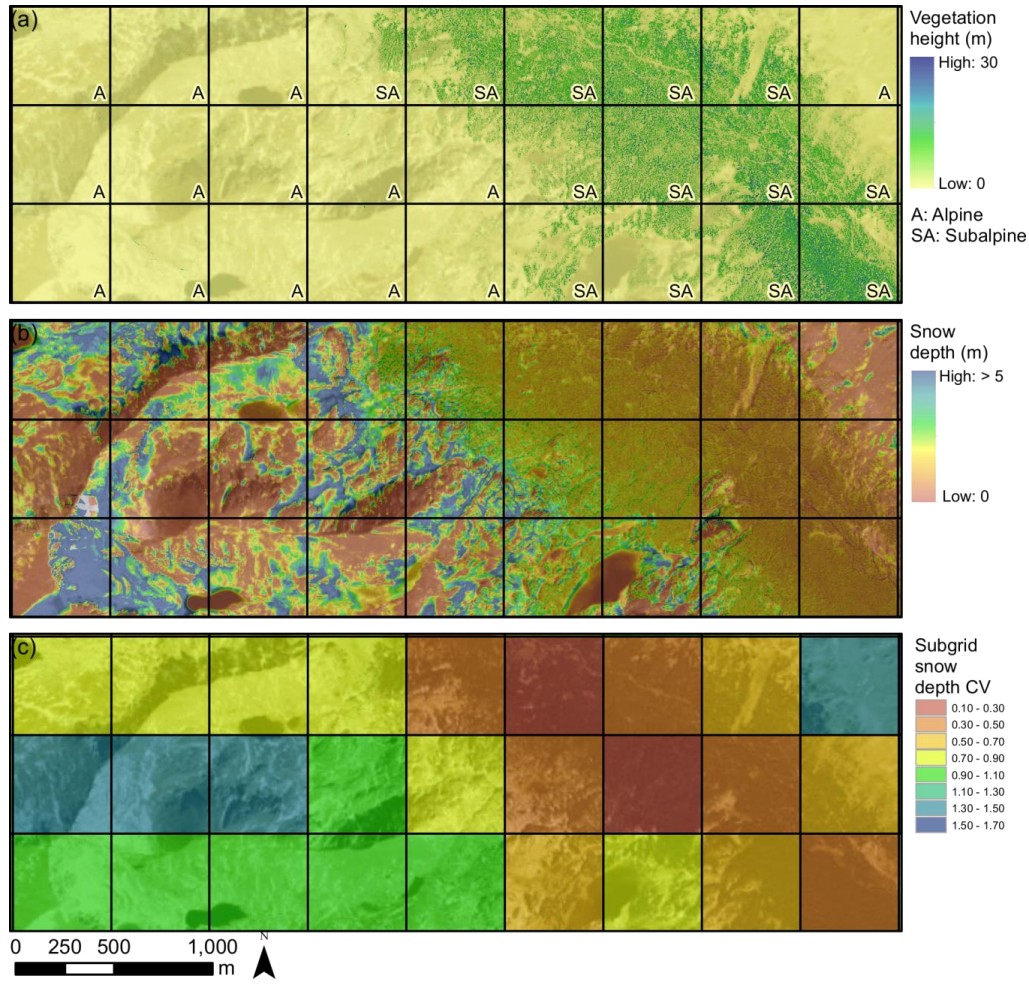

**Figure 6: Close up map of selected study grids showing the distribution of (a) vegetation height and ecosystem type, (b) snow depth, and (c) subgrid $CV_{ds}$ value. Area of close up is highlighted in Figure 1.**



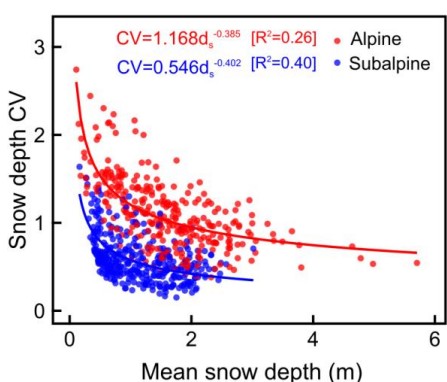

**Figure 7: Nonlinear relation of $CV_{ds}$ and $d_s$ for alpine (red) and subalpine (blue) study grids (500 m resolution).**





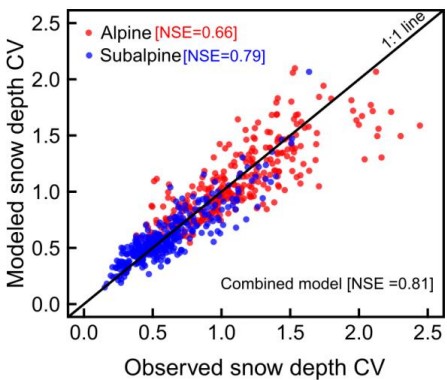

Figure 8: Modeled versus observed $CV_{ds}$ for the alpine (red) and subalpine (blue) multiple linear regression models.



**Table 1: Bivariate correlations (Pearson's *r* coefficient) between snow depth coefficient of variation ($CV_{ds}$) and the mean and standard deviation ($\sigma$) of snow depth ($d_s$), vegetation height (*VH*) and coefficient of variation of vegetation height ($CV_{VH}$), Normalized Difference Vegetation Index (*NDVI*), canopy density (*CD*), elevation (*z*), slope (*S*), winter clear-sky incoming solar radiation ($Q_{sw\downarrow}$), maximum upwind slope (*Sx*), and topographic position index (*TPI*) for both alpine and subalpine study grids. Correlations are also shown for the residuals from the detrended nonlinear relation of $CVd_s$ and $d_s$. Bold values represent statistical significance (*p* value < 0.05).**

| | $CV_{ds}$ (alpine) | $CV_{ds}$ (subalpine) | $CV_{ds}$ (alpine) $d_s$ residuals | $CV_{ds}$ (subalpine) $d_s$ residuals |
|---|---|---|---|---|
| $d_s$ | **-0.60** | **-0.45** | --- | --- |
| $\sigma_{ds}$ | -0.06 | **0.25** | --- | --- |
| *VH* | **-0.38** | **-0.48** | **-0.28** | **-0.71** |
| $\sigma_{VH}$ | **-0.38** | **-0.57** | **-0.24** | **-0.59** |
| $CV_{VH}$ | **0.16** | **0.28** | -0.08 | **0.61** |
| *NDVI* | **0.2** | -0.10 | **-0.13** | **-0.42** |
| $\sigma_{NDVI}$ | **-0.12** | **0.25** | -0.01 | **0.55** |
| *CD* | -0.06 | **-0.32** | **-0.21** | **-0.64** |
| $\sigma_{CD}$ | -0.06 | **0.30** | **-0.26** | **0.50** |
| *z* | **0.17** | **-0.22** | **0.32** | **0.18** |
| $\sigma_z$ | -0.07 | 0.09 | **0.16** | **0.29** |
| *S* | -0.03 | 0.06 | **0.25** | **0.28** |
| $\sigma_S$ | -0.06 | 0.13 | **0.37** | **0.38** |
| $Q_{sw\downarrow}$ | 0.10 | -0.02 | -0.07 | **-0.17** |
| $\sigma_{Qsw\downarrow}$ | -0.07 | -0.03 | **0.21** | **0.21** |
| *Sx* | 0.02 | 0.08 | **0.29** | 0.09 |
| $\sigma_{Sx}$ | 0.07 | 0.10 | **0.43** | **0.28** |
| *TPI* | **0.28** | **0.11** | **0.15** | 0.04 |
| $\sigma_{TPI}$ | -0.09 | 0.09 | **0.29** | **0.33** |





**Table 2: Multiple linear regression equation variables and coefficients of the alpine and subalpine $CV_{ds}$ models. The multiple linear regression is of the form:** $y = \beta_0 + \beta_1 x_1 + \beta_2 x_2 + \ldots + \beta_n x_n$ **where $y$ is the dependent variable, $x_1$ through $x_n$ are $n$ independent variables, $\beta_0$ is the regression intercept, and $\beta_1$ through $\beta_n$ are $n$ regression coefficients. Units of the model variables area as following: snow depth (m), maximum upwind slope (°), clear-sky solar radiation (W m$^{-2}$), canopy density (%), surface slope (°).**

|  | Alpine model | Subalpine model |
|---|---|---|
| $Y$ | $\log(CV_{ds})$ | $CV_{ds}^{0.5}$ |
| $\beta_0$ | 9.00E-03 | 8.45E-01 |
| $\beta_1$ | -1.02E+00 | -2.84E-01 |
| $x_1$ | $d_s^{0.5}$ | $\log(d_s)$ |
| $\beta_2$ | 1.00E-02 | -9.79E-05 |
| $x_2$ | $Sx$ | $CD^2$ |
| $\beta_3$ | 3.42E-01 | 1.12E-02 |
| $x_3$ | $\log(\sigma_{Sx})$ | $\sigma_S$ |
| $\beta_4$ | 1.84E-03 | --- |
| $x_4$ | $Q_{SW\downarrow}$ | --- |