# Peer review of "Subgrid snow depth coefficient of variation within complex mountainous terrain"

_The Cryosphere, 2016_

## Referee Comment (RC1) · Anonymous Referee #1 · 23 Oct 2016

Major comments: 1. One major issue with this paper is that the content addressing the 3rd objective stated in the introduction is far from enough. If they would like to develop a general methodology for parameterizing CVds in complex mountainous terrain, the author should test their methods on other mountainous areas, such as the Sierra Nevada (Opentopograhy does have a snow-on snow-off lidar dataset, and also Airborne Snow Observatory (NASA JPL) collected three years of time-series lidar data in two basins of the Sierra). But the only testing the author addressed was cross-validation, which is quite site-specific and did not provide enough evidence this method could be applied on other regions. The authors should shrink down this broader goal in the current manuscript.

2. The second issue is that I expected model selection to be a key point being addressed for a statistical modeling paper, but it is not there. The author should elaborate

a lot more in detail about model selection so the reader will know how your final model was selected. And are model-selection results site-specific? Your process of selecting the right model could be great information to other researchers who are working on similar topics but in different geographic areas.

Minor comments:

Page 4, line 3-8: Although 10-m cell have been shown to influence snow-depth distributions, were spatial resolutions in previous studies the same as what you are looking at (1 m)? And how do you justify the ds statistics calculated from the 1-m resolution data could be parameterized by 10-m resolution topographic and vegetation data using statistical models?

Page 4, line 24: Why use these three window sizes and what are their effects on the TPI image?

Page 4, line 25-30: Why not use the canopy data from the lidar? And how do you account for the forest changes over time.

Page 5, line 13-14: I am really confused about this sentence. Rephrase for clarity.

Page 5, line 28-30: If the influence of ds on CVds was removed using a best-fit power function, why does Table 1 still have Âňds's Pearson's r coefficient? Are these results in Table 1 before detrending or after detrending?

Page 5, line 34-35: The reason that mean z was excluded in model testing is not convincing. As it is known that ds is highly dependent on z, why not exclude ds as well? And also why is VH not commonly available? If you have snow-on and snow-off lidar for the area you are looking at, why you do not have VH? And if you are trying to create a parameterization of CVds globally for all mountainous regions, then failing to test your models in other regions is a serious shortcoming. So from my understanding, your model is already site specific so why not explore as many variables as you could? Page 6, line 2-4: I think that under Gaussian linear regression, Mallows' Cp is the

same as AIC. And also why you are using AIC over likelihood-ratio test if your models are nested? Is it necessary to penalize the number of parameters?

Page 6, line 10-13: The way you presented k-fold cross-validation is incorrect. You should shuffle your dataset first and then iterate over the k-folds. Page 6, section 3.1: Although snow conditions are close for the two lidar campaigns, you should not blend them together for modeling, especially then there is a precipitation event between the two. Because the spatial snow conditions are not completely independent of time. Therefore, temporal effects on the snowpack will be a confounding factor for modeling the CVds when you mixing the data from two dates together. And all the causal inferences you made and regression model parameters you estimated will be confounded by temporal effects.

Page 7, line 15: What is your U-statistic for the Mann-Whitney test?

Page 8, line 3-4: What transformations were applied on $\sigma$_sx and CD?

Page 8, section 3.4: What is your justification of the variables used in the multiple linear regression model? And where is the result of your model selection using AIC?

Page 16, Could you also show the footprint of snow-off lidar data you used?

Page 17, Similar comment as before. You should not say the spatial distribution of the two dates' lidar flight are the same.

———————————————————

---

## Short Comment (SC1) · 1 Nov 2016

The authors used a lidar-derived snow depth dataset to investigate the subgrid variability of snow in complex mountainous terrain. They found that the snow depth coefficient of variation is well correlated with snow depth, topography and vegetation conditions/metrics, and can be parameterized by these factors/parameters. The results are interesting. However, recent studies also showed that deposition of light-absorbing aerosols (mainly black carbon and dust) exerts significant impacts on snow properties over mountains, which reduces snow albedo and hence accelerates snow melting (e.g., Painter et al., 2013; He et al., 2014; Liou et al., 2014; Lee et al., 2016). How would the heterogeneous distribution of light-absorbing aerosols affect the variability of snow? Would the parameterization be improved if the effect of light-absorbing aerosol deposition were included? I suggest adding some discussions on this aspect as well

as those recent studies, which could be helpful in understanding potential uncertainty associated with the estimate of snow variability in mountainous regions.

References:

He, C., Li, Q. B., Liou, K. N., Takano, Y., Gu, Y., Qi, L., Mao, Y. H., and Leung, L. R.: Black carbon radiative forcing over the Tibetan Plateau, Geophys. Res. Lett., 41, 7806–7813, doi:10.1002/2014gl062191, 2014.

Lee, W.-L., K. N. Liou, C. He, H.-C. Liang, T.-C. Wang, Q. Li, Z. Liu, and Q. Yue: Impact of absorbing aerosol deposition on snow albedo reduction over the southern Tibetan plateau based on satellite observations, Theoretical and Applied Climatology, 1-10, doi:10.1007/s00704-016-1860-4, 2016.

Liou, K. N., Takano, Y., He, C., Yang, P., Leung, L. R., Gu, Y., and Lee, W. L.: Stochastic parameterization for light absorption by internally mixed BC/dust in snow grains for application to climate models, J. Geophys. Res.-Atmos., 119, 7616–7632, doi:10.1002/2014jd021665, 2014.

Painter, T. H., M. G. Flanner, G. Kaser, B. Marzeion, R. A. VanCuren, and W. Abdalati (2013), End of the Little Ice Age in the Alps forced by industrial black carbon, Proc. Natl. Acad. Sci. U.S.A., 110(38), 15,216–15,221, doi:10.1073/pnas.1302570110.

---

## Referee Comment (RC2) · Anonymous Referee #2 · 7 Nov 2016

Comments:

The manuscript by Sexstone et al. sets out to examine how the coefficient of variability (CV) of snow depth varies over differing grid resolutions in their study area of the Front Ranges of the Colorado Rockies, based on a high resolution lidar-derived snow depth dataset obtained there. The study includes the additional objectives to evaluate how topographic and vegetation conditions influence the variability of CV at the sub-grid level, and to develop a methodology for parameterizing CV over complex mountain terrain. However, in the end, the study makes very little advancement in the field of snow hydrology and modelling, with most of the results serving only as a limited empirical case study, and it fails to deliver on its final objective (the parameterization of CV), which would have been its only real major new contribution. To fix this, I believe, would be beyond the scope of major revision, requiring substantial new analyses and a fundamentally different approach. For these reasons I feel the paper should be rejected at this time.

With regard to the first objective (to determine the range of CV values that observed within varying grid resolutions throughout the study area), there is little purpose in examining the variation of CV of snow depth over varying grid resolutions. If the intention is to develop a means of parameterizing CV for mountain terrain, then the first step is to focus on objectively chosen landscape units (hydrological response units, grouped response units) over which to examine CV, or alternatively, use the information from the lidar snow depths to examine how the landscape could be disaggregated so as to minimize CV within the groups. This would make any parameterization more robust and potentially applicable beyond the limited conditions observed in this study. Making use of additional data from other sites or other years/seasons would greatly add to the value of this exercise.

As for the second objective (to evaluate the effects of mean snow depth, forest, and topography characteristics on subgrid CV), there is little here that is fundamentally new, and not enough of an advancement to warrant publication. Indeed the authors rightly point out that future research could investigate how the variability of snow depth varies across different geographic regions, snow regimes, snow seasons (particularly high and low snow years), and over time in a single season. This, together with the suggestions above, are what would make a more meaningful contribution. As it stands, the results add very little to what is already understood about snow accumulation in complex mountain terrain.

Finally, the third objective (to develop a methodology for parameterizing CV within complex mountainous terrain) is not achieved in this study. The manuscript describes an empirical study of the relationship between topographic and vegetation conditions for a single locale at a single point in time. It is not physically based (i.e. in the sense of utilizing known physics of snow accumulation, redistribution, and ablation based on meteorological conditions during the winter and spring), there is no basis for predicting

[Figure]

CV outside of this area and time, and it offers little more than what is already known: that snow depth and its variability can be statistically related to physical and biological landscape elements.
* * *

---

## Referee Comment (RC3) · Anonymous Referee #3 · 22 Nov 2016

Mountainous snowpacks accumulate deep snow and generally exhibit a large range in depth variability over short distances. Subgrid snow distribution is important for reliably simulating energy and mass exchanges between the land and atmosphere in snow covered mountainous regions. The authors attempt to use the snow depth coefficient of variation (CVds), as a metric of subgrid snow variability within complex mountainous terrain, as they feel that the current range of published CVds in this environment is quite variable and is not well parameterized. The objectives of the paper are to use high resolution LiDAR snow depth estimates recorded in the Front Range Mountains of north-central Colorado, over a 321 km2 study site to:

1) Determine the range of CVds values that are observed within varying grid resolutions throughout the study area

2) Evaluate the effects of mean snow depth, forest and topography characteristics on subgrid CVds

3) Develop a methodology for parameterizing CVds within complex mountainous terrain.

The authors fall short on achieving their objectives and/or provide evidence that their findings have already been well documented in previous publications.

1) The authors document that the range in mean CVds values decreases with increasing grid resolutions, which is not a novel result. They do point out that at 500 m there is very little difference in the range of CVds values with increasing grid resolutions >500 m and <1000 m. Their results also highlight a broad range in CVds values, which are as large (between 10th and 90th percentiles) as those currently presented in the literature and documented by Clark et al., (2011) – See specific comment PG 7 – Line 16-17 for details. The differing ranges and median CVds values for both alpine (non-forested) and subalpine (forested) mountainous environments also closely match those presented in Clark et al., (2011).

2) The authors evaluate the effects of mean snow depth, forest and topography characteristics on subgrid CVds values. They find that snow depth was the most important driver of CVds variability in both alpine and subalpine areas which are consistent with previous studies (PG 8 Line 32), which is not a novel result. The strong correlations between CVds (snow distribution) and vegetation parameters, wind exposure, and topography are also not new results, and have been previously well documented.

3) The authors develop two simple statistical models (alpine/subalpine) for predicting subgrid CVds from mean snow depth and topography/canopy information that can be used for parameterizing CVds at this site. However, as the authors state on PG 10 – Lines 1-10, analysis of subgrid snow variability across a greater geographical region (not limited to a single site), with differing snow regimes, using multiple years of data would be needed to improve the applicability of this parameterization for complex

mountainous terrain in general, which was the original 3rd objective .

The paper is well written and presented. The idea of using continuous, high resolution LiDAR data to better parameterize the CVds values in any environment is exciting, compared to the difficult and challenging task of conducting discrete in situ snow surveys. There is always some uncertainty associated with whether or not the discrete measurements will adequately capture the full range of CVds across the study domain. It is promising that the range of LiDAR derived CVds values observed in this study domain seem to match the results of many different in situ snow survey campaigns presented in Clark et al., (2011). It is also promising that the median LiDAR CVds values from this site agree well with the values defined by Liston (2004) for global modeling applications. These positive results highlight that past/present in situ snow survey methods seem to be adequately capturing the CVds values for mountainous environments. However, as the authors themselves highlight, LiDAR is a relatively new (and expensive) technology, and therefore existing LiDAR data are limited to small study domains, and typically of only a single snap-shot for a single winter season, or a very short multi-year time series. Therefore, developing methodology for parameterizing LiDAR derived CVds values that would be applicable to global snow classification using LiDAR data is not quite feasible yet.

Due to the lack of novel results, and the reasons discussed above, I cannot recommend publishing this paper in its current form. Major revisions, including a re-defining of objectives would be necessary. I feel that this paper should be rejected at this time.

Specific Comments:

PG 2 - Line 35-36: Please provide references for the common 1000 m resolution hydrologic and land surface models

PG 4 – Lines 24: TPI is calculated using three different scales (30 m, 110 m, 210 m – All three of these scales are relatively small – why did you choose these three options?). Typically, combining of small and large scale TPI allows for different landforms classes

to be identified. Was this completed? Or were only the raw TPI values used in this analysis? Only a single TPI correlation is presented in Table 1. Which scale of TPI was used? How different were the correlations between scales?

PG 4 – Lines 25-30: Move the description of the canopy density dataset (Line 28-30) to Line 25 when you state that additional forest canopy spatial datasets were also used. Please state who the author/source of the canopy density data is and what year they produced the map (produced by the Multi-Resolution Land Characteristics (MRLC) Consortium, based primarily on Landsat imagery from 2001-2011)

PG 4-5 – Lines 34-2: Awkward sentence. Please re-phrase or split into two sentences.

PG 5 – Line 2-3: Suggest changing to ' This study attempts to evaluate the subgrid variability of ds at a comparable grid resolution to a 1000 m grid resolution of an operational snow model (Carroll et al., 2006).'

PG 5 – Lines 3-4: I recommend that you add the description of developing the categorical variables alpine and subalpine (Lines 19-23) to the beginning of the site description to help explain the coloring and extent of the grids in Figure 1, otherwise when you first look at Figure 1 it is not clear on why the grids do not cover the entire LiDAR extent.

PG 5 – Line 5: How and why did you choose these particular grid resolutions to test (100, 200, 300, 400, 500, 750, 1000?)

PG 5 – Lines 31-36: If you deemed some variables unsuitable for parameterizing the snow depth CV and were excluded from the model testing, then simply remove all mention of them from the text (results and methods sections).

PG 6 – Line 19: Suggest changing to 'increase in subgrid snow depth variability'

I would also suggest using a plot of change in snow depth data if available, rather than the niveograph of SWE. SWE may stay the same if density increases due to melt events (melt/re-freeze processes). Whereas snow depth always decreases during melt events. This paper is also concerned with parameterizing the snow depth coefficient of

variation. (See attached Figure 1. taken from Dingman, 2002)

Reference: Dingman SL. 2002. Physical Hydrology Second Edition. Prentice Hall: Upper Saddle River, New Jersey

PG 6 – Line 21: Suggest changing to: 'snowpack melt conditions'

PG 7 – Line 5: Suggest changing to ' tended to increase with increasing grid size for all alpine study grids,'

PG 7 – Line 6: The 500 m resolution study grids had sample size of n=642 stated in the text. In Figure 1, the sample size is n=650.

PG 7 – Line 7: how did you define comparable? What thresholds did you use?

PG 7 – Line 11: insert reference to Figure 5 after providing median snow depth values.

PG 7 – Line 16-17: The snow depth CV values for mid-latitude tree-less mountains presented in Clark et al., 2011 Figure 2 (> 0.50 to < 1.5) agree with the range of snow depth CV values observed in this study for the alpine grids (0.61 to 1.57). The reported range of snow depth CV values for the subalpine (forested) grids (0.3 to 0.98) also fall within the range of reported Clark et al., 2011 values for the mid-latitude mountainous forest (0.10 to <1.0). Watson et al 2006 reported some very large CV values compared to all other studies, exceeding the max subalpine values found in this study (~1.75), however, the majority of their reported CV values were <1.0 agreeing with the results of this study. Therefore the range of observed CV values reported in this study seem to be already well documented for complex mountainous terrain.

PG 8 – Line 22-29: Liston's 2004 definition of nine different snow distribution categories and their typical snow depth CV values appear to agree well with the median CV values reported in this within this study. The reported range of CV values found within Clark et al., 2011 agree well with the values found within this study site (See comment for PG 7 – Line 16-17 above), and therefore it appears that CV values in complex mountainous terrain may already be adequately documented.

PG 9 – Line 5-9: Agreed. These two sentences describe future work that would be necessary to achieve objective 3 of this paper.

PG 10 – Line 7-10: Agreed. Future studies with multiple years of LiDAR measurements would be useful for achieving objective 3.
* * *
[Figure]

Dingman 2002, Figure 5-24

**Fig. 1.**

---

## Editor Comment (EC1) · R. D. Brown (Editor) · 23 Nov 2016

Dear authors, thank you for submitting your paper to The Cryosphere. After reading the review comments and considering the recommendations of the reviewers, I regret to inform you that I cannot recommend your paper for publication in The Cryosphere. The main issues are the lack of significant new findings, and the inability of the paper to develop a physically-based approach for estimating CV. I trust the comments are helpful in rethinking the methodology as the reviewers saw merit in the paper's overall goal of applying Lidar to understand snow depth variability over complex terrain.

Best regards, Ross Brown (ed.)
* * *